# Effect of PAS-LuxR Family Regulators on the Secondary Metabolism of *Streptomyces*

**DOI:** 10.3390/antibiotics11121783

**Published:** 2022-12-08

**Authors:** Naifan Zhang, Yao Dong, Hongli Zhou, Hao Cui

**Affiliations:** 1College of Chemistry and Pharmaceutical Engineering, Jilin Institute of Chemical Technology, Jilin 132022, China; 2College of Biology & Food Engineering, Jilin Institute of Chemical Technology, Jilin 132022, China; 3Engineering Research Center for Agricultural Resources and Comprehensive Utilization of Jilin Province, Jilin Institute of Chemical Technology, Jilin 132022, China

**Keywords:** *Streptomyces*, secondary metabolism, PAS-LuxR, regulator

## Abstract

With the development of sequencing technology and further scientific research, an increasing number of biosynthetic gene clusters associated with secondary *Streptomyces* metabolites have been identified and characterized. The encoded genes of a family of regulators designated as PAS-LuxR are gradually being discovered in some biosynthetic gene clusters of polyene macrolide, aminoglycoside, and amino acid analogues. PAS-LuxR family regulators affect secondary *Streptomyces* metabolites by interacting with other family regulators to regulate the transcription of the target genes in the gene cluster. This paper provides a review of the structure, function, regulatory mechanism, and application of these regulators to provide more information on the regulation of secondary metabolite biosynthesis in *Streptomyces,* and promote the application of PAS-LuxR family regulators in industrial breeding and other directions.

## 1. Introduction

Secondary *Streptomyces* metabolites appear in the late stages of growth, when nutrients are insufficient and growth rates are reduced; they provide more than half of all antibiotics in clinical use [1]. The secondary metabolism of *Streptomyces* is a complex process that is closely linked to the complex differentiation of cells. Thus, various regulatory approaches are used for the metabolic process in *Streptomyces* in which different regulators interact and form a large and complex regulatory network that is rare in prokaryotes. Depending on the source, the different regulatory factors of a bacterium can be divided into internal and external (e.g., environmental) factors. Environmental factors include carbon, nitrogen, and phosphate, which are essential nutrients for the primary metabolism of the bacterium and serve as regulatory substances on the secondary metabolism of the bacterium. Regulatory substances from the bacterium itself typically include small molecules, such as highly phosphorylated guanosine acids (ppGpp), γ-butyrolactones, small proteins such as σ-factors, and regulatory proteins encoded by regulatory genes’ biosynthetic gene clusters.

The regulatory genes in the biosynthesis of secondary metabolites in *Streptomyces* can be divided into those that contain only a DNA binding domain (DBD) and those that have an additional signal recognition or energy conversion region in addition to the DBD. Most of the regulators in *Streptomyces* contain an additional region [2]. According to the range of regulation, most regulators are pathway-specific, while a few regulators are pleiotropic regulators. Depending on the structure and function of the regulatory proteins, they can be classified as tetracycline repression (TetR), xenobiotic response element (XRE), *Streptomyces* antibiotic regulatory protein (SARP), multiple antibiotic resistance regulator (MarR), large ATP-binding regulators of the LuxR (LAL), PAS-LuxR, and other families [3]. More than 100 TetR family regulators have been identified, rendering it the largest family of regulators available [4]. XRE family regulators are the second most common family and are commonly found in bacteria such as *Serratia marcescens*, *Streptomyces*, and *Pseudomonas aeruginosa* [5]. SARP regulators were the first family to be identified and studied, but they are only found in *Actinomycetes* and mostly in *Streptomyces* [6]. MarR family regulators are commonly found in the genomes of *Streptomyces* with an average of 50 MarR family regulators per genome [7], and only one conserved structural domain was found for binding to DNA in its structure [6]. The LAL family is a large class of regulators that bind to ATP, typically consisting of 900–1000 amino acids. The PAS-LuxR family, a late class of regulators, combines an N-terminal PAS domain with a C-terminal helix–turn–helix (HTH) motif of the LuxR type. PAS domain, a signal module that monitors changes in light, redox potential, oxygen energy level of a cell, and small ligands was first found in eukaryotes and was named after homology to the Drosophila period protein (Per), the aryl hydrocarbon receptor nuclear translocator protein (ARNT), and the Drosophila single-minded protein (Sim) [8]. Similar to other families, the PAS-LuxR family regulators control secondary metabolic processes by influencing the transcription of functional genes. However, considering the presence of PAS, this regulator may play a regulatory role by sensing extracellular environmental changes. In addition, in terms of regulatory mode, the binding site of PAS-LuxR regulators contains a 16-bp palindrome-like region and adjusts to the consensus CTVGGGAWWTCCCBAG (V represents A, C, or G; W stands for A or T; B stands for C, G, or T), which is located in the −35 region in promoters of the regulated genes [9].

This paper provides a review of the structure, function, regulatory mechanism, and application of PAS-LuxR family regulators to improve the study of secondary metabolite biosynthetic regulation in *Streptomyces* and promote the application of PAS-LuxR family regulators in industrial breeding and other directions. PAS-LuxR family, regulator, and secondary metabolism were selected as the keywords for the search in Google Scholar from 2007 to 2022.

## 2. PAS-LuxR Family Regulators

The number of PAS-LuxR family regulators is relatively small, and the 18 reported regulators are listed in Table 1. The regulators of this family in different strains with the conserved regulatory function can be exchangeable. The genes encoding PAS-LuxR regulators can be found in all known polyene macrolide biosynthesis gene clusters [10] and in the two nonpolyene products pathways of coronafacoyl-L-isoleucine (CFA-L-Ile) and wuyiencin. The structures of these products are shown in Figure 1.

### 2.1. PAS-LuxR Family Regulators Are Present in the Type I Polyketide Synthase (PKS) Pathway

As shown in Figure 2, PAS-LuxR regulatory genes are predominantly found in the PKS pathway. For example, the most well-reported biosynthetic gene cluster of pimaricin (also known as natamycin) has five PKS genes (*pimS0*, *pimS1*, *pimS2*, *pimS3*, and *pimS4*), three oxidoreductase genes (*pimD*, *pimF*, and *pimG*), three glycosylation genes (*pimC*, *pimJ*, and *pimK*), three transporter genes (*pimA*, *pimB*, and *pimH*), one pleiotropic effect regulatory gene (*pimT*) [9], one thioester hydrolase gene (*pimI*), and one sterol oxidase gene (*pimE*) [11,12]. In addition to the PKS and postmodifying genes, two regulatory genes, namely, *pimM* and *pimR*, encode a PAS-LuxR family regulator and a SARP-LAL family regulator. Reedsmycin is a nonglycosylated polyene macrolide, and its gene cluster includes four PKS genes (*rdmG*, *rdmH*, *rdmI,* and *rdmJ*), a resistance protein gene (*rdmK*), a PaaI family thioesterase gene (*rdmM*), an acyl carrier protein (ACP) gene (*rdmN*), an acyl coenzyme A synthase gene (*rdmO*), two genes with the methyltransferase gene (*rdmB*), the ferritin gene (*rdmL*), and five regulatory protein genes (*rdmA*, *rdmC*, *rdmD*, *rdmE*, and *rdmF*) that encode a PAS-LuxR family protein [13]. The PAS-LuxR family regulators are also present in the biosynthetic gene clusters of all polyene macrolides, including nystatin, filipin, tetramycin, aureofuscin, candicidin/FR-008, and amphotericin. CFA-L-Ile is an amino acid analogue, and the CFA-L-Ile gene cluster consists of two PKS genes (*cfa6* and *cfa7*), one acyl carrier protein (ACP) gene (*cfa1*), two acyl coenzyme A ligase genes (*cfa5* and *cfal*), one oxidoreductase gene (*scab79681*), a cytochrome P450 monooxygenase gene (*sabc79691*), two reductase genes (*cfa8* and *sabc79721*), a hydroxybutyryl coenzyme A dehydrogenase gene (*sabc79711*), and two regulatory genes (*orf1* and *cfaR*, which encode a PAS-LuxR family protein) [14]. Wuyiencin is an aminoglycoside antibiotic [15,16] that was mainly studied in biocontrol. The gene cluster has only been partially sequenced and it contains PAS-LuxR gene *wysR*. PAS-LuxR family regulators are mainly present in the PKS pathway.

### 2.2. Structure of PAS-LuxR Family Regulators

PAS-LuxR family regulators are a class of proteins containing a PAS-sensing structural domain at the N-terminal and a DBD region at the C-terminal with HTH [17]. PAS monitors light and oxygen changes in redox potential, and cellular energy and is a component of histidine and serine/threonine kinases, chemoreceptors, and photoreceptors that are involved in tropism, biological clock, and ion channels [8]. Unlike other sensing elements, proteins containing the PAS region are located in the cytosol and can directly sense signals inside the cell, while the above proteins can cross the cell membrane to sense signals from the external environment. The DBD region at the C-terminus of the PAS-LuxR family regulators is similar to that of LAL family regulators and is also derived from HTH-LuxR. The PAS-LuxR family regulators are highly conserved and generally consist of 152–243 amino acids, as shown in Figure 3. In its primary structure, the conserved region of PAS region is generally present in 1–130 amino acids, and the conserved amino acid residues are AX LDXXLTIXXANX_n_DXCGRXFXDXHPSVXQPLXRQFXXLXE. All the HTH structural domains are generally located at 117–221 amino acids, and their conserved amino acid residues are LX_3_DARILEGIAAGXSTXPLAXXLYLSRQGVEYHVTXLLRKLXVPNRAAL. In the secondary structure shown in Figure 4, all PAS domains contain the PAS core, helical connector, and β-scaffold. PAS core has the highest density of conserved residues in the PAS domain and is the key core region of PAS structure. The helical connector and β-scaffold are less conserved than the PAS core is [18]. The β-scaffold forms a long platform with a characteristic β-sheet that supports the PAS core. As illustrated in Figure 5, each regulator contains at least one HTH structure.

### 2.3. Developmental Tree Analysis

The phylogenetic tree was constructed using the neighbour-joining algorithm, and the bootstrap test (Bootstrap) was resampled for 500 times to increase the confidence value of the evolutionary tree. All the PAS-LuxR family regulators were derived from the same ancestor. The overall similarity of the PAS-LuxR family regulatory amino acid sequences was high, but the sequences that generate the same sequences of the same antibiotic still differed.

## 3. Regulation of Related Product Biosynthesis by PAS-LuxR Family Regulators

PAS-LuxR family regulators play a regulatory role together with the regulators of other families. According to the product structure classification, they can be mainly divided into polyene and nonpolyene macrolides, which are most studied in detail in pimaricin, nystatin, filipin, tetramycin, candicidin (also known as FR008), amphotericin, NppA1, and CFA-L-Ile. However, the regulatory mechanisms of reedsmycin and wuyiencin need to be further determined.

### 3.1. Regulation of Polyene Macrolide Biosynthesis by PAS-LuxR Family Regulators

#### 3.1.1. Vertical Regulatory System Formed with SARP-LAL and PAS-LuxR Family Regulators

The PAS-LuxR family of regulators was first reported in the pimaricin pathway; a tetraene macrolide produced by *S. natalensis* ATCC 27448 is also produced by *S. chattanoogensis*, *S. lydicus*, and *S. gilvosporeus* [19]. The pimaricin pathway in *S. natalensis* involves two positive regulators, namely, the PAS-LuxR family regulator PimM and the SARP-LAL family regulator PimR. PimM affects the biosynthesis of pimaricin by regulating the transcription of PKS genes (*pimS0*, *pimS1*, *pimS2*, *pimS3*, and *pimS4*), oxidoreductase genes (*pimD*, *pimF*, and *pimG*), glycosylation genes (*pimC*, *pimJ*, and *pimK*), transporter genes (*pimA* and *pimB*), and the thioester hydrolase gene (*pimI*). However, *pimE*, *pimH*, *pimM*, and *pimR* are not regulated by PimM. PimR affects pimaricin biosynthesis indirectly by regulating *pimM* through binding to the site of TGGCAAGAAAGCGGCAGGTGTTCGGCAAG with the heptameric repeats in the promoter of *pimM*. In combination with the regulatory role of PimM, a vertical regulatory system is formed between the two regulators [20]. The pimaricin biosynthetic gene cluster in *S. chattanoogensis* has 17 open reading frames, including two regulatory genes, namely, *scnRI* and *scnRII*, which correspond to *pimR* and *pimM* with sequence identities of 85% and 96%, respectively [21,22,23]. ScnRII also belongs to the PAS-LuxR family. The analysis of ScnRII (192aa) revealed that it has 99% sequence identity with the positive regulator AurJ3M of *S. aureofuscus* and 96% amino acid similarity with PimM. Genes *scnA*, *scnE*, *scnD*, and *scnRII* are partially regulated by ScnRI, while *scnS1* and *scnT* are not regulated by ScnRI. ScnRII affects pimaricin biosynthesis by regulating eight genes, namely, *scnA*, *scnE*, *scnD*, *scnI*, *scnJ*, *scnK*, *scnS1*, and *scnS2*, which bind to the promoters of the above genes and recruit RNA polymerase to initiate their transcription, forming a vertical regulatory system [24]. Considering the difference in the target genes regulated by PimM and ScnRII, the amino acid sequences of these two regulators were analyzed independently. The results reveal that 186aa out of 192aa are identical. The six amino acid residues that did not match appeared in the PAS structural domain, and the genes within the two gene clusters were not identical, which may account for their different regulatory mechanisms. The SlnM in pimaricin producer *S. lydicus* A02 is a member of the PAS-LuxR family [25]. The identities of *slnM* with *scnRII* and *pimM* genes are both 96%. However, the identity of SlnM with ScnRII was higher than that with PimM or SgnRII, as indicated by the closer relationship between SlnM and ScnRII in Figure 6. The biosynthetic gene clusters of pimaricin have not been reported in *S. gilvosporeus* [26]. 

Filipin is a 28-membered cyclopentenyl macrolide produced by *S. avermitilis*, *S. filipinensis*, and *S. miharaensis* [27,28,29], of which the most detailed biosynthetic process has been reported in *S. avermitilis*. Similar to pimaricin, the filipin pathway has two different regulators, namely, the SARP-LAL family regulator PteR and the PAS-LuxR family regulator PteF. PteR is homologous to PimR, while *pteF* is homologous to PimM. The transcription of all PKS genes (*pteA1*, *pteA2*, *pteA3*, *pteA4*, and *pteA5*) are significantly reduced in the *pteF*-disrupted strain (Δ*pteF*), in which the expression of *pteA3*, *pteA4*, and *pteA5* significantly differed, suggesting that *pteF* has a direct or indirect regulatory effect on these genes [30]. The expression of *pteB*, *pteC*, and *pteD* is also reduced, and the downregulated expression of *pteB* and *pteC* was consistent with those of *pteA3*, *pteA4,* and *pteA5*. Therefore, they were cotranscribed as *pteA3-A4-A5*-*B*-*C*, and the promoter of *pteA3* was regulated by PteF. By contrast, *pteR*, *pteH*, and *pteG* expression levels were upregulated in Δ*pteF* compared with those in wild-type strain, suggesting that PteF may be a negative regulator or that the disruption of PteF may activate the negative regulator of the genes. For the two other filipin producers, FilF- FilR and PteF-PteR correspond to the regulator combinations of PimM-PimR. SARP-LAL family regulator PimR can bind to heptameric repeats separated by four nucleotides, TGGCAAGAAAGCGGCAGGTGTTCGGCAAG, at least two protein monomers are required for effective binding, and the bound *pimM* region does not overlap with the −35 box [31]. Considering the potential site of TGGCAAGAAAGCGGCAGGTGTTCGGCAAG in the promoters of *petF* and *filF*, the vertical regulatory system formed with SARP-LAL and PAS-LuxR family regulators in the filipin pathway. 

#### 3.1.2. Network Regulatory System Formed with LAL and PAS-LuxR Family Regulators

In comparison with the vertical regulatory system, the PAS-LuxR family regulator may form a network regulatory system with the LAL family regulators in polyene macrolides pathway, such as tetramycin, nystatin, amphotericin, candicidin, and NPP. Tetramycin consists of A and B components and is a 26-membered tetraene macrolide produced by *S. ahygroscopicus* and *S. hygrospinosus*, respectively [32,33]. The tetramycin pathway in *S. ahygroscopicus* contains three LAL family regulators (TtmRI, TtmRII, and TtmRIII) and a PAS-LuxR family regulator (TtmRIV). The glycosylation genes (*ttmK* and *ttmC*) and PKS genes (*ttmS0*, *ttmS1*, *ttmS2*, *ttmS3*, and *ttmS4*) are fully regulated by TtmRIV. The sterol oxidase gene (*ttmE*), oxidoreductase genes (*ttmG* and *ttmF*), glycosylation gene (*ttmJ*), transporter genes (*ttmA* and *ttmB*), and precursor synthesis gene (*ttmP*) are partially regulated by TtmRIV. The *ttmL*, *ttmD*, *ttmRIV*, *ttmRIII*, *ttmRII* and *ttmRI* genes are not regulated by TtmRIV. TtmRIV affects the biosynthesis of tetramycin by regulating the transcription of the 14 genes above. TtmRI can directly regulate the transcription of *ttmK*, *ttmC*, *ttmG*, *ttmF*, *ttmS0*, *ttmS1*, *ttmRIV*, *ttmRIII*, *ttmRII*, *ttmRI*, *ttmJ*, *ttmA*, *ttmB*, and *ttmP* and indirectly regulate the transcription of *ttmE*, *ttmS2*, *ttmS3*, and *ttmS4*. TtmRII directly regulates the transcription of *ttmG*, *ttmF*, *ttmS0*, *ttmRIV*, *ttmRIII*, *ttmRII*, and *ttmRI* and indirectly regulate the transcription of *ttmE*, *ttmK*, *ttmC*, *ttmS1*, *ttmS2*, *ttmS3*, *ttmS4*, *ttmJ*, *ttmA*, *ttmB*, and *ttmP*. TtmRIII directly regulates the transcription of *ttmE*, *ttmG*, *ttmF*, *ttmS0*, *ttmS1*, *ttmRIV*, *ttmRIII*, *ttmRII*, and *ttmRI* and indirectly regulate the transcription of *ttmK*, *ttmC*, *ttmS2*, *ttmS3*, *ttmS4*, *ttmJ*, *ttmA*, *ttmB*, and *ttmP*. Hence, a network regulatory system formed with TtmRI, TtmRII, TtmRIII, and TtmRIV regulates the biosynthesis of tetramycin [34]. 

The pathway of the 38-membered tetraene macrolide nystatin in *S. noursei* has four regulators, namely, LAL family regulators NysRI, NysRII, and NysRIII, and PAS-LuxR family regulator NysRIV. The transcription of the *nysH* gene, which is responsible for transport, is not regulated by NysRI, but rather by NysRII, NysRIII, and NysRIV [35]. The transcription of glycosylation genes *nysDI* and *nysDII* is not regulated by NysR. The transcription of PKS genes *nysA* and *nysI* genes is regulated by NysR. The transcription of *nysRI* and *nysRIV* is regulated by all the four regulators, although the degree of regulation differs. The transcription of *nysRI* is moderately regulated by NysRIII, NysRIV, and NysRI and fully regulated by NysRII. The transcription of *nysRIV* is moderately regulated by NysRIV and fully regulated by NysRI, NysRII, and NysRIII [35]. The regulation of NysR to the structural genes and the regulatory genes indicated that they form a network regulatory system. 

The LAL and PAS-LuxR family regulators are also identified in the candicidin, NppB1, and salinomycin pathway. Candicidin is an heptane macrolide that is produced by *S.* sp. FR-008 [36]. The candicidin biosynthetic gene cluster involves 21 genes, and four regulatory genes encode the PAS-LuxR family regulator FscRI and the LAL family regulators FscRII, FscRIII, and FscRIV. In Δ*fscRI*, the RT-PCR results show that the transcriptions of the six PKS genes (*fscA*, *fscB*, *fscC*, *fscD*, *fscE*, and *fscF*), the glycosylation genes (*fscMI*, *fscMII*, and *fscMIII*), and the translocation genes (*fscTI* and *fscTII*) were reduced compared with the wild-type strain, suggesting that FscRI regulates the biosynthesis of candicidin by affecting the transcription of all the above genes. FscRII and FscRIII belong to the top of the regulatory network in which FscRII negatively regulates the expression of FscRIII and positively regulates the expression of FscRIV. FscRIII has a positive regulatory effect on FscRI and FscRII. Considering that FscRI can unidirectionally compensate for the deletion of FscRIII and FscRIV, and FscRIV can unidirectionally compensate for the deletion of FscRI, FscRIV can partially replace FscRII, while FscRI can partially replace FscRIII and FscRIV [37]. In addition to the abovementioned genes regulated by *fscRI*, the transcript of both *fscRI* and the precursor synthesis gene *pabC* decreased in Δ*fscRIV*, suggesting that *fscRI* and *fscRIV* have direct or indirect regulatory effects on the above genes and they regulate one another [38]. Therefore, the biosynthesis of candicidin is regulated by the regulatory network formed with the four regulators. 

In addition, some type I PKS biosynthetic pathways contain several LAL and PAS-LuxR family regulators, and the relationship between their regulatory roles has not been fully determined. A heptaene macrolide amphotericin is produced by *S. nodosus* and is commonly used clinically. Four regulators were also identified in the amphotericin pathway, including the three LAL family regulators AmphRI, AmphRII, AmphRIII, and PAS-LuxR family regulator AmphRIV. Tetraene macrolide NPPA1 produced by *Pseudonocardia autotrophica* is 300-fold more water-soluble and 10-fold less hemolytic than nystatin A1 is, but its antimicrobial activity is 50-fold lower [39]. The increased antimicrobial activity NppB1 was obtained with the inactivation of the enolase structure in the fifth module of its biosynthetic gene *nppC*, but its yield is less 10% than that of NppA1 [40]. The biosynthetic gene cluster contains six regulatory genes, namely, *nppRI*, *nppRII*, *nppRIII*, *nppRIV*, *nppRV*, and *nppRVI*, of which *nppRI*, *nppRIII*, and *nppRV* encode LAL family regulators, and *nppRIV* encodes a PAS-LuxR family regulator. The sequence identities of NppRIV with *nysRIV* and *pimM* are 45.6% and 45%, respectively [41]. Two LAL family regulators encode genes, *salRI* and *salRII*, and PAS-luxR family regulator encode gene *salRIII*, which is also present in the biosynthetic gene cluster of the polyether ion carrier salinomycin produced by *S. albus* CCM4719. The sequence identities of *salRIII* with *pteF*, *fscRI*, *nysRIV*, and *amphRIV* are 44%, 47%, 48%, and 50%, respectively [42]. PF1 and PF2 are polyene macrolides produced by *S. marokkonensis* M10, and its structure is similar to that of candicidin. The regulators in the PF1 and PF2 pathway correspond to those of candicidin pathway, a PAS-LuxR family regulator MarRI to FscRI (98% identity), three LAL family regulators MarRII to FscRII (99% identity), MarRIII to FscRIII (97% identity), and MarRIV to FscRIV (97% identity) [43,44]. Although the regulatory mechanisms of PAS-LuxR family and LAL family regulators in the above PKS pathway have not been reported, on the basis of the reported regulatory networks in the biosynthetic pathway of the same kind of substances, several of their regulators also play a similar regulatory role.

#### 3.1.3. Regulatory System Formed with PAS-LuxR and Other Family Regulators 

In addition to the regulatory system in Section 3.1.1 and Section 3.1.2, other families’ regulators act in conjunction with PAS-LuxR family regulators. For example, the aureofuscin pathway in *S. aureofuscus* has two regulators, namely, PAS-LuxR family regulator AurJ3M and RhtB family regulator AurT [45,46]. In the *aurJ3M* gene deletion strain, RT-PCR results showed that *aurB*, *aurC*, *aurG*, *aurF*, *aurS0*, *aurS1*, and *aurD* are not transcribed, the transcript levels of *aurI*, *aurJ*, and *aurA* genes are reduced compared with the wild-type strain, and almost no effect was observed on *aurR*, *aurE*, *aurH*, and *aurJ3M* genes. Therefore, the deletion of *aurJ3M* has no effect on the transcription of genes located downstream of AurJ3M, and AurJ3M regulates the biosynthesis of aureofuscin by affecting the transcription of the PKS gene *aurS0.* The amino acid transporter protein AurT regulates the biosynthesis of aureofuscin by affecting the secretion of PI factors [11], whose structural analogs activate the expression of related genes by binding to specific receptors within the gene cluster [47]. Hence, the production of aureofuscin in *S. aureofuscus* is coregulated by two regulators, namely, AurJ3M and AurT, from different transcription factor families. Reedsmycin is a nonglycosylated polyene macrolide produced by the marine *Streptomyces* of *S.* sp. CHQ-64. The antifungal activity of reedsmycin is comparable to that of nystatin in the inhibition of *Candida albicans*. The reedsmycin pathway has five regulators, namely the XRE family regulator RdmA, LuxR family regulators RdmC, RdmD, RdmE, and PAS-LuxR family regulator RdmF [48]. The regulatory mechanism of RdmF has not been reported.

**Table 1 antibiotics-11-01783-t001:** Information regarding the reported PAS-LuxR family regulators.

Regulator	*Streptomyces*	Size (aa)	Product	Type of Product	Target Genes	Yield in Overexpression Strain	Reference
PimM ScnRII SgnRII SlnM	*S. natalensis* (*S.chattanoogensis*) (*S.gilvosporeus*) (*S. lydicus*)	192	Pimaricin	PM	*pimS0*, *pimS1*, *pimB*, *pimC*, *pimD*, *pimF*, *pimG*, *pimI*, *pimJ*, *pimS2*, *pimS3*, *pimS4*, *pimA*, *pimK* *scnA*, *scnE*, *scnD*, *scnI*, *scnJ*, *scnK*, *scnS1*, *scnS2*	240–460%	[11,12,19]
RdmF	*S.* sp. CHQ-64	233	Reedsmycins	NG-PM	NR	250%	[13]
CfaR	*S. scabiei*	152	CFA-L-Ile	AAA	*cfa1*(27kb, from *cfa1* to *scab79721*)	1000%	[14]
WysR	*S. wuyiensis* CK-15	193	Wuyiencin	AG	*wysE*, *wysRI*, *wysRIII*		[15]
PteF FilF	*S. avermitilis* (*S. filipinensis*) (*S. miharaensis)*	192	Filipin	PM	*pteA1*, *pteA2*, *pteA3*, *pteA4*, *pteA5*, *pteB*, *pteC*, *pteD*, *pteH*, *pteG*, *pteR*	NR	[27,28,29]
TtmRIV TetrRIV	*S. ahygroscopicus* (*S. hygrospinosus*)	201	Tetramycin	PM	*ttmK*, *ttmC*, *ttmS0*, *ttmS1*, *ttmS2*, *ttmS3*, *ttmS4*, *ttmE*, *ttmG*, *ttmF*, *ttmJ*, *ttmA*, *ttmB*, *ttmP*	333% (Tetramycin A)	[32,33]
NysRIV	*S. noursei*	210	Nystatin	PM	*nysH*, *nysA*, *nysI*, *nysRI*, *nysRIV*	NR	[35]
FscRI	*S.* sp. FR-008	222	Candicidin	PM	*fscA*, *fscB*, *fscC*, *fscD*, *fscE*, *fscF*, *fscMI*, *fscMII*, *fscMIII*, *fscTI*, *fscTII*	NR	[36]
AmphRIV	*S. nodosus*	243	Amphotericin	PM	NR	400% (in Δ*amphNM*)	[39]
NppRIV	*P. autotrophica*	213	NPPA1	PM	NR	−50%	[40]
SalRIII	*S. albus*	231	Salinomycin	PIC	NR	NR	[42]
MarRI	*S. marokkonensis* M10	148	PF	PM	NR	NR	[43]
AurJ3M	*S.aureofuscus*	192	Aureofuscin	PM	*aurB*, *aurC*, *aurG*, *aurF*, *aurS0*, *aurS1*, *aurD*, *aurI*, *aurJ*, *aurA*	600%	[45]

PM, polyene macrolide; AAA, amino acid analogues; NG-PM, nonglycosylated polyene macrolide; AG, aminoglycoside; PIC, polyether ionic carrier; NR, not reported.

### 3.2. Regulation of the Nonpolyene Macrolides Biosynthesis by PAS-LuxR Family Regulators 

CFA-L-Ile, a phytotoxin, is produced by *S. scabies* and causes the development of potato blast. The biosynthetic gene cluster of CFA-L-Ile contains 15 genes, and 13 of them are cotranscribed in a multicistron, while the two remaining genes, namely, PAS-LuxR family regulator gene *cfaR* and partial ThiF family gene *orf1*, are transcribed in another transcript [14]. The *cfaR* deletion strain has no or low-level transcripts for *cfa5*, *cfa6*, and *cfl* compared with the wild-type strain. The cotranscribed gene *orf1* of *cfaR* is not affected in any way, indicating that *cfaR* activates the transcription of approximately 27 kb transcripts from *cfa1* to *scab79721*, but it does not regulate itself. EMSA results show that CfaR only binds to the *cfa1* promoter region, further demonstrating that CfaR regulates the CFA-L-Ile pathway directly through a vertical regulatory system [49].

Wuyiencin is an aminoglycoside antibiotic produced by *S.wuyiensis* CK-15, and a PAS-LuxR gene *wysR* is present in the biosynthetic gene cluster. Wuyiencin is widely used for fungal control in crops and has low toxicity. The sequencing analysis of WysR (193 amino acids) revealed 84.27% identity with NysRIV and more than 60% identity with other PAS-LuxR family regulatory protein sequences. The transcript levels of *wysE* (thioesterase), *wysRI* (LAL family regulator), and *wysRIII* (DeoR family regulator) were analyzed in the wuyiencin biosynthetic gene cluster, and the results show that the transcript levels of *wysRIII* and *wysE* are remarkably lower in *wysR* deletion strain than that in the wild-type strain, while the transcript levels of *wysRI* remarkably increased compared with wild-type strains [16]. In the *wysR* overexpressing strain, the transcript levels of *wysRIII* and *wysE* remarkably increased, whereas the transcript levels of *wysRI* are significantly reduced compared with the wild-type strain [15]. Therefore, WysR directly or indirectly regulates the expression of other potential regulatory genes such as *wysRI* and *wysRIII* to affect the transcription of early wuyiencin PKS genes and regulates *wysE* to affect the late biosynthetic enzymes in wuyiencin production [16]. Moreover, WysR regulates wuyiencin biosynthesis and morphological development by influencing possible regulatory genes such as *bld*, *whi*, *chp*, *rd1*, and *ram* family genes, indicating its crucial role in the morphological development [15].

### 3.3. Regulatory Mechanisms of PAS-LuxR Family Regulators 

As the prototype of a PAS-LuxR family regulator, the regulatory mechanism of PimM in pimaricin biosynthesis has been widely studied. PimM affects the transcription of target genes by binding to their promoters [50]. A comprehensive study on PAS-LuxR regulators has been conducted in the biosynthesis of polyene macrolides of pimaricin, nystatin, amphotericin, and filipin. Results show that the expression of *amphRIV*, *nysRIV*, or *pteF* in *pimM* deletion strains can restore the production of pimaricin. The expression of *pimM* in *S. nodosus* and *S. avermitilis* can increase the yield of amphotericin and filipin, respectively. The expression of *pimM* in *pteF* deletion strain can also restore the production of filipin [30]. GST-PimM could bind with the promoters of controlled biosynthetic genes of amphotericin (*amphA*, *amphDI*, *amphI, amphH*, and *amphDIII*), nystatin (*nysA*, *nysDI*, *nysI*, *nysH*, and *nysDIII*), filipin (*pteA1*), and pimaricin (*pimK*, *pimS2*, *pimI*, *pimJ*, *pimA*, *pimE*, *pimS1*, and *pimD*). The binding site of the PAS-LuxR class regulator is located in a 16-bp palindromic conserved sequence 5′-CTVGGGAWWTCCCBAG-3′ (V stands for A, C, or G; W stands for A or T; B stands for C, G or T) [50]. Therefore, PAS-LuxR family proteins are regulators with relatively conserved regulatory functions.

The relatively conserved function of PAS-LuxR family regulators can be attributed to their regulatory role across gene clusters and antibiotic varieties. FscRI can regulate both candicidin and antimycins across gene clusters in *S. albus* S4. In addition, FscRI is required for the heterologous expression of antimycin biosynthetic gene clusters in *S. coelicolor*. Notably, the 16 bp binding sites of PAS-LuxR family regulator exist in the promoters of *antB2*, *antB1*, and *antC*, which are important genes in antimycin biosynthesis [51]. The increment of *pimM* copies increased the production of candicidin and antimycin at the same time in *S. albus* J1074 [52].

Following the above characteristic sequences, the potential binding sites of unspecified PAS-LuxR family regulators were searched, and the results show similar binding sites in the promoters of *nysDIII*, *nysH*, *nysI*, *nysA*, *nysRI*, and *nysRIV* in the biosynthetic gene cluster of nystatin in *S. noursei* ATCC 11455, the promoters of *amphH*, *amphI*, *amphL*, *amphM*, *amphDI*, *amphA*, and *amphB* in the biosynthetic gene cluster of amphotericin in *S. nodosus*, the promoters of *scnK*, *scnS4*, *scnS3*, *scnS2*, *scnI*, *scnJ*, *scnA*, *scnB*, *scnE*, *scnC*, *scnF*, *scnS0*, *scnS1*, and *scnD* in the biosynthetic gene cluster of pimaricin in *S. chattanoogensis* L10, the promoters of *rdmA*, *rdmC*, *rdmF*, *rdmG*, *rdmH*, *rdmI*, *rdmK*, *rdmL*, and *rdmN* in the biosynthetic gene cluster of reedsmycins in S.sp. strain OUC6819, and the promoters of *pteA1* and *pteA2* in the biosynthetic gene cluster of filipin in *S. avermitilis*. The sequence logo is shown in Figure 7. Therefore, the PAS-LuxR family regulators derived from different secondary metabolites in different *Streptomyces* are functionally conserved and can be substituted for each other. 

However, these findings were all obtained from different strains. In *S. ahygroscopicus*, although TtmRIV is present together with NysRIV, when *ttmRIV* is deleted, tetramycin is no longer produced when nystatin production is enhanced [34]. Similarly, when *nysRIV* is deleted, nystatin is no longer produced, and tetramycin production is enhanced (unpublished results). Therefore, NysRIV and TtmRIV cannot achieve functional substitution but can exhibit independent and precise regulation in *S. ahygroscopicus*. Therefore, although TtmRIV and NysRIV belong to the same PAS-LuxR family regulators, their regulatory mechanisms differ. On the basis of the comparison of the amino acid sequences of TtmRIV with other PAS-LuxR family regulators, amino acids at 25–43 aa at the N-terminus of the PAS region of TtmRIV had 19 additional amino acid residues compared with the corresponding positions of other regulators (Figure 8), the same observation was made in the LuxR at the C-terminus. On the basis of the comparison of the secondary structures, the PAS regions of AmphRIV, FscRI, NysRIV, and PimM all contain three α-helices and two β-folded structures in the PAS region, while the PAS region of TtmRIV contains three α-helices and one β-folded structure. Structurally, TtmRIV do not form another β-fold structure because of the 19 additional amino acid residues, indicating that the PAS structure in TtmRIV differs from the normal PAS structure that has been reported. As shown in Figure 9, on the basis of the comparison of the conserved sequence 5′-YVSGGAWWTCCSBR-3′ of the TtmRIV binding sites sequence with the conserved sequence 5’-CTVGGGAWWTCCCBAG-3′ of the PAS-LuxR family regulator binding site sequence reported (Y for C or T; V for A, C or G; S for C or G; W for A or T; B for C, G or T; R for A or G), difference was observed at the −5/4 and −7/6 bases. In *S. avermitilis*, PteF formed vertical regulatory system with SARP-LAL family regulator in the filipin pathway. In addition, PteF affected the transcriptions of genes related to various metabolic processes, including genetic information processing; DNA, energy, carbohydrate, and lipid metabolism; morphological differentiation; and transcriptional regulation, among others, but were particularly related to 10 potential secondary metabolites [53]. Therefore, the study on the regulatory mechanism of PAS-LuxR family regulators is still incomplete, thus requiring further studies on the presence of two PAS-LuxR family regulators in the same strain without functional crossover for the understanding of the precise regulation of such family regulators. 

## 4. Application of PAS-LuxR Family Regulators

PAS-LuxR family regulators influence the biosynthesis of target products by regulating the transcription of target genes. On the basis of this feature, the production of metabolites can be changed by the over-expression of PAS-LuxR family regulator genes in the biosynthetic gene cluster or knocking out them in the non-target gene cluster. 

As shown in Table 1, the production of pimaricin is completely eliminated by blocking *pimM* in *S. natalensis*. After *pimM* complementation, pimaricin production could be restored to the wild-type level, and PimM overexpression could reach 2.4 times of the wild-type level. Upon the overexpression of ScnRII in *S. chattanoogensis*, compared with the wild-type strain, the yield of pimaricin on YSG medium is 3.3 times higher and that of pimaricin on YEME medium is 4.6 times higher. When *slnM* is overexpressed in *S.lydicus*, the production rates of pimaricin in the overexpressing strains containing ermEp* promoter, native promoter, and ermEp*- native promoter are 2.4, 1.9, and 3 times higher than those in the wild-type strain, respectively, on the YEME medium without sucrose. On the YSG medium, *slnM* overexpressing strains containing ermEp*-native promoter produced 2.1 times more pimaricin than wild-type strains do [25].

*S. ahygroscopicus* produces two polyene macrolides, namely, tetramycin and nystatin. In the *ttmRIV* deletion strain, nystatin production increased 2.1 times, and a high-production nystatin strain was obtained [46]. When the *ttmRIV* gene was overexpressed, the yield of tetramycin A was 3.33 times that of the wild-type strain, but the yield of tetramycin B did not change, because the *ttmD* gene, which is responsible for the transformation of tetramycin A to tetramycin B, is not regulated by TtmRIV. When *nysRIV* was disrupted, the yield of nystatin was completely eliminated, and the yield of tetramycin increased by 1.28 times. In *S. aureofuscus* SYAU0709, blocking AurJ3M resulted in stopped the production of aureofuscin, and the overexpression of AurJ3M could increase the production of aureofuscin by six times [54]. The *pteF* deletion mutant resulted in a 38% loss of filipin yield and delayed spore formation in the starting strain. The complementation of *pteF* restored the yield to wild-type levels and restored the rate of spore formation. The heterologous complementation of *pimM* to the deletion mutant strain of *pteF* fully restored filipin to wild-type levels. The overexpression of the *amphRIV* gene in the *S. nodosus* Δ*amphNM* mutant strain resulted in an approximately three-fold increase in the yield of the related derivative [55]. The loss of *rdmF* also completely deprives the yield of reedsmycin, in which the level was almost restored to wild-type levels after complementation, and the overexpression of *rdmF* resulted in 2.5 times increase in reedsmycin yield [33]. PAS-LuxR transcriptional activators share the similarity regulatory pattern. By contrast, the overexpression of NppRIV in the *P. autotrophica* reduces NPP production [40]. This phenomenon may be related to the different regulatory networks composed of six regulators present in *P. autotrophica*. 

For nonpolyene products, neither Δ*cfaR*Δ*PAS* nor Δ*cfaR*Δ*LuxR* produces CFA-L-Ile in *S. scabies*. *cfaR+orf1* overexpression strains produced more than 10 times the yield of *cfaR* overexpression strains, whereas the overexpression of *orf1* alone did not remarkably affect the yield of CFA-L-Ile [49]. *wysR* gene deletion resulted in the complete disappearance of wuyiencin yield. Subsequently, the wild-type level was restored after complementation, and the overexpression of *wysR* resulted in an increase in wuyiencin yield with a value that was thrice that of the wild-type yield [15].

Aside from improving the product yield and species, PAS-LuxR family regulators are also used to regulate the heterologous expression of gene clusters. For example, the heterologous expression of antimycin biosynthetic gene clusters in *S. coelicolor* is required to simultaneously express FscRI and thus activate the antimycin biosynthetic gene clusters. The heterologous expression of AmphRIV, NysRIV, or PteF can restore the production of pimaricin in *pimM* deletion strains. The expression of PimM can increase the yield of amphotericin and filipin in *S. nodosus* and *S. avermitilis*, respectively. The expression of *pimM* in *pteF* deletion strain also restored the production of filipin [30]. These findings broaden the potential applications of PAS-LuxR family regulators, such as the activation of silenced gene clusters to obtain new products and the regulation of gene expression from different sources in combinatorial biosynthesis. 

## 5. Conclusions

PAS-LuxR family regulators are a class of regulators that have recently been discovered, and most of them exist in the biosynthetic pathway of PKS type I products. They regulate the transcription of the target gene by binding to the characteristic 5′-CTVGGGAWWTCCCBAG-3′ sequence on the promoters of target genes. Except for PteF, which indirectly affects the morphological differentiation, all other regulators are pathway-specific. When playing regulatory roles, PAS-LuxR family regulators need to cooperate with other family regulators, mainly including the SARP-LAL-PAS-LuxR vertical regulatory system and LAL-PAS-LuxR network regulatory system. In the regulatory system, the PAS-LuxR regulators locate in the core position and have the potential role of corresponding extracellular environment changes, which increase the complexity and accuracy of the regulatory system to ensure the normal metabolism of *Streptomyces*. The PAS-LuxR family regulator has a similar structure and function that can realize cross-strain inter-replacement. Although the regulatory mechanism of PAS-LuxR family regulator has been widely studied, PAS-LuxR family regulators TtmRIV and NysRIV play independent regulatory functions in *S. ahygroscopicus*, which is rare, and the precise regulatory mechanism is not clear. Therefore, the regulators of this family need to be further studied. The precise regulation of TtmRIV and NysRIV is the stability factor that maintains *S. ahygroscopicus* metabolism. Further studies on the independent regulation of PAS-LuxR family regulators in *Streptomyces* would contribute to a deeper understanding of polyene macrolides biosynthesis of regulation, and more comprehensive understanding of the precise regulation of gene expression in *Streptomyces*. This study also provides a theoretical foundation for the use of precise regulation to synthesis of new antibiotics by synthetic biology means in the efficiently and directly. Lastly, it may play a positive role in promoting the research and development of new antibiotics.

## Figures and Tables

**Figure 1 antibiotics-11-01783-f001:**
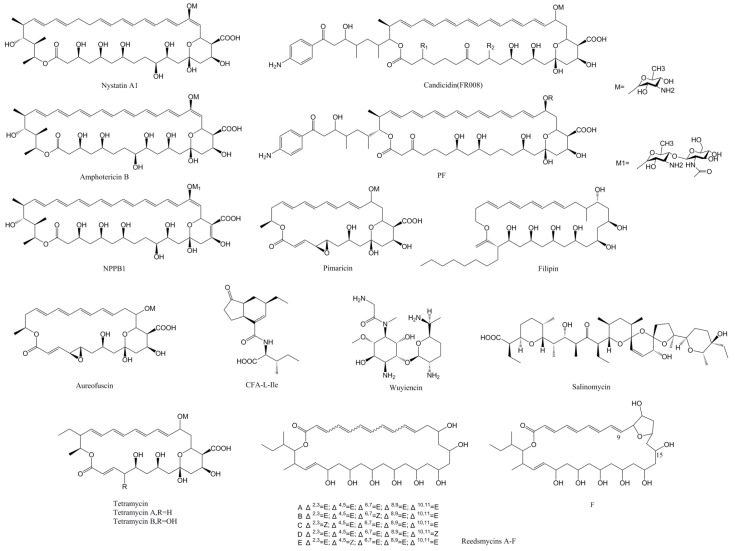
Structures of secondary metabolites regulated by the PAS-LuxR family regulators.

**Figure 2 antibiotics-11-01783-f002:**
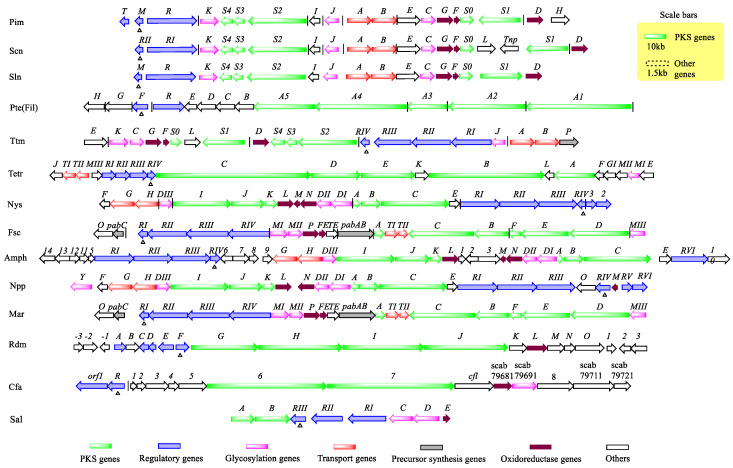
Biosynthetic gene clusters containing the PAS-LuxR genes. Sln is a biosynthetic gene cluster that produces pimaricin in *S.gilvosporeus*. Scn is a biosynthetic gene cluster that produces pimaricin in *S. chattanoogensis*. Pim is a biosynthetic gene cluster that produces pimaricin from *S. natalensis*. Ttm is a biosynthetic gene cluster that produces tetramycin from *S. ahygroscopicus*. Tetr is a biosynthetic gene cluster that produces tetramycin from *S. hygrospinosus*. Nys is a biosynthetic gene cluster for the production of nystatin by *S. noursei*, Amph is a biosynthetic gene cluster that produces amphotericin from *S. nodosus*. Fsc is a biosynthetic gene cluster for the production of candicidin by *S.* sp. FR-008. Pte (Fil) is a biosynthetic gene cluster producing filipin from *S. avermitilis* (*S. filipinensis*). Cfa is a biosynthetic gene cluster for the production of CFA-L-Ile by *S. scabiei*. Rdm is a biosynthetic gene cluster that produces reedsmycins from *S.* sp. CHQ-64. Npp is a biosynthetic gene cluster that produces NPPA1 from *P. autotrophica*. Mar is a biosynthetic gene cluster that produces PF from *S. marokkonensis* M10. Sal is a biosynthetic gene cluster that produces salinomycin from *S. albus*. The symbol of Δ and ❙ represent PAS-LuxR regulators coding genes and binding sites.

**Figure 3 antibiotics-11-01783-f003:**
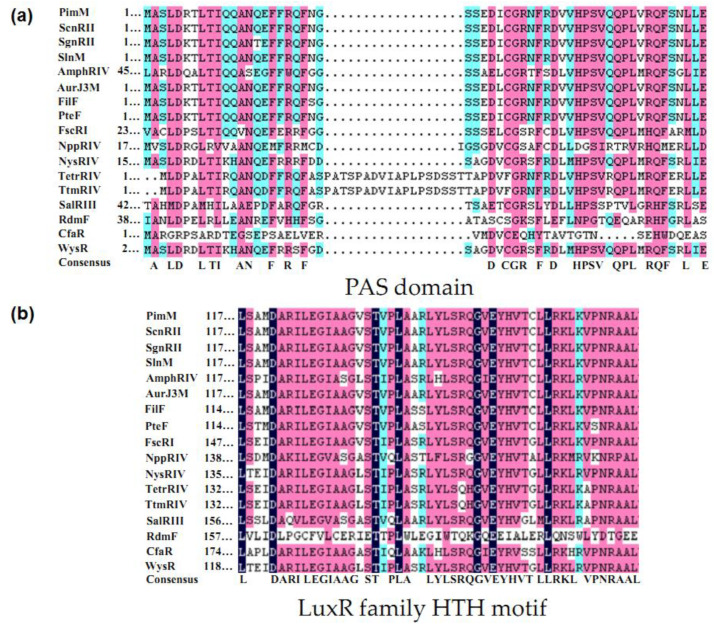
Sequence comparison of PAS-LuxR family regulatory factors. (**a**) PAS domain; (**b**) LuxR family HTH motif. X is an undefined amino acid residue.

**Figure 4 antibiotics-11-01783-f004:**
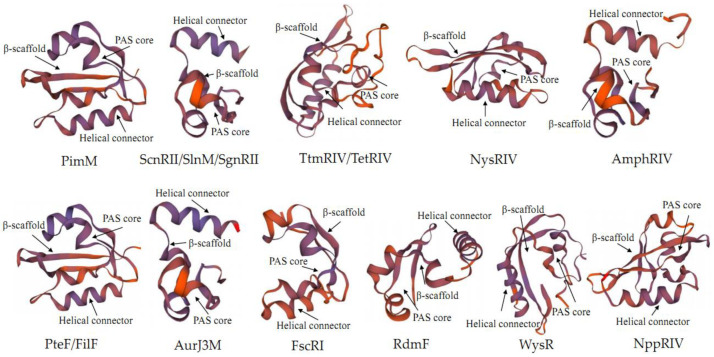
Simulated secondary structures of N-terminal PAS structure of PAS-LuxR family regulators by SWISS-MODEL on https://swissmodel.expasy.org/, accessed on 25 July 2022.

**Figure 5 antibiotics-11-01783-f005:**
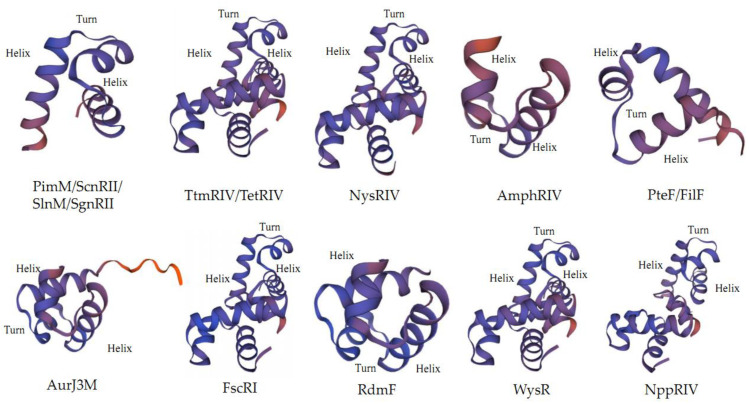
Simulated secondary structures of C-terminal HTH motif of PAS-LuxR family regulators by SWISS-MODEL on https://swissmodel.expasy.org/, accessed on 25 July 2022.

**Figure 6 antibiotics-11-01783-f006:**
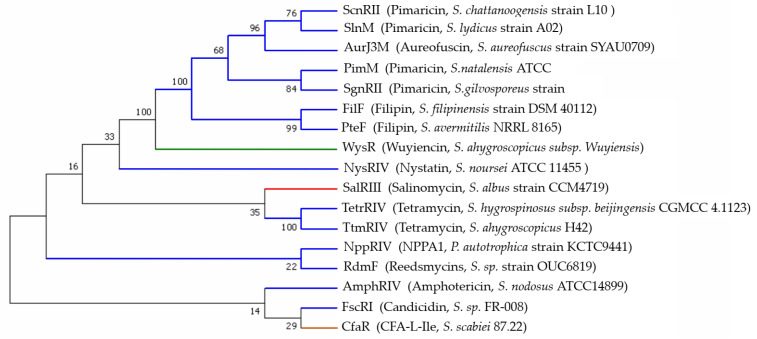
Phylogenetic tree of PAS-LuxR family regulators.

**Figure 7 antibiotics-11-01783-f007:**
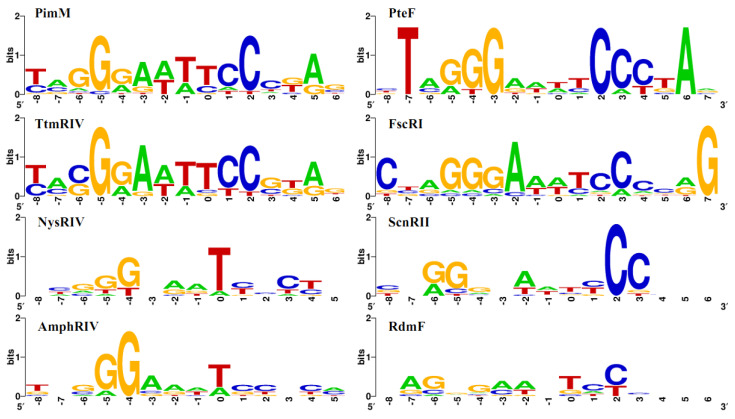
Prediction of binding sites of the PAS-LuxR family regulators.

**Figure 8 antibiotics-11-01783-f008:**
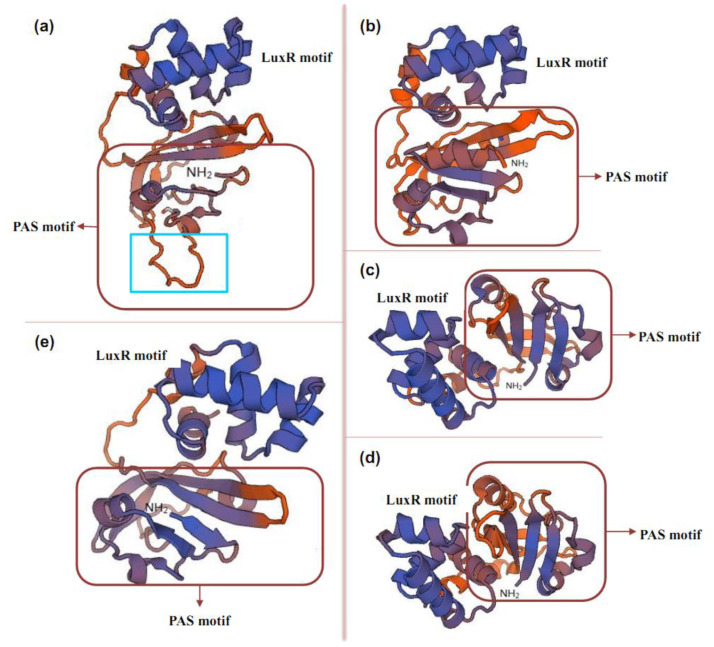
Comparison of amino acid sequences in the PAS region. (**a**), TtmRIV; (**b**), FscRI; (**c**), AmphRIV; (**d**), NysRIV; (**e**), PimM.

**Figure 9 antibiotics-11-01783-f009:**
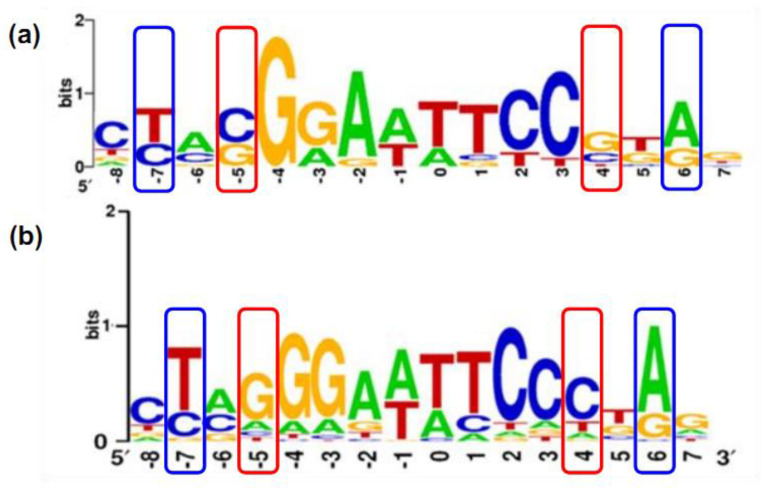
Sequence logo comparison of TtmRIV and PAS-LuxR family regulon binding sites. (**a**) Sequence logo of TtmRIV binding sites. (**b**) Sequence logo of PAS-LuxR family regulator binding sites reported in the literature.

## Data Availability

Not applicable.

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
