# Peer review of "Effect of PAS-LuxR Family Regulators on the Secondary Metabolism of Streptomyces"

_antibiotics, 2022, doi:10.3390/antibiotics11121783_

Round 1

Reviewer 1 Report

In this Review Paper, Zhang et al. summarize and update knowledge regarding the PAS-LuxR-type transcriptional regulator distributed to the biosynthetic gene cluster for secondary metabolism in Streptomyces. Streptomyces, the Gram-positive filamentous soil bacteria, is renowned for its ability to produces a wide variety of biologically active compounds as secondary metabolites. The expression of the genes involved in the biosynthesis of those compounds is developmentally regulated associating with that of the genes involved in the cell differentiation. Generally, the biosynthesis gene cluster contains a gene(s) encoding a transcriptional regulator, which specifically controls the transcription of the cluster. The transcriptional regulators are grouped into several types. Of these, the so-called SARP (Streptomyces antibiotic regulatory protein) family has been well studied for its role and application to the development of hyper-producer strain. Similarly, other types of regulatory proteins are also characterized for their role and function. This review dealing with the knowledge of the PAS-LuxR type regulators is welcome and expected to provide useful information to the readers of the journal. This reviewer has the following comments.

Lines 38-39

In this sentence, the authors introduce ppGpp, gamma-butyrolactone, and sigma factor as the regulator of secondary metabolism in Streptomyces, but it is somewhat jumbling. It is better to separate the description regarding small molecules (ppGpp and gamma-butyrolactones) from that of sigma factors (transcriptional regulatory proteins).

Line 64

Provide basic information for PAS-LuxR family, including what PAS and LuxR stand for and how this type of regulator controls the transcription of the target genes (molecular mechanism of regulation). The authors provide related information in lines 121-127, but a clear explanation should be added in the Introduction. It would be useful if the authors introduced the knowledge from detailed studies on the role of this type of transcriptional regulator. What is the difference of PAS-LuxR from other types of regulators? This is an important point for the authors to shed light on this type of regulator.

Figure 2

It would be useful if the localization of the promoter(s) controlled by the PAS-LuxR-type regulator is indicated in this figure. The meaning of the three different scale bars (upper, right) was not clear to this reviewer.

Line 143

There is no indication of corresponding panel designations (a and b) in Fig. 3.

Line 255

The reference(s) should be cited in this position.

Lines 342-353

The description of this position needs effective citation. 

English grammar and expression should be edited throughout the text.

Author Response

Response to Reviewer 1 Comments

Comment 1: Lines 38-39: In this sentence, the authors introduce ppGpp, gamma-butyrolactone, and sigma factor as the regulator of secondary metabolism in Streptomyces, but it is somewhat jumbling. It is better to separate the description regarding small molecules (ppGpp and gamma-butyrolactones) from that of sigma factors (transcriptional regulatory proteins).

Response 1: The sentence in lines 38-39 had been revised to be “Regulatory substances from the bacterium itself typically include small molecules, such as highly phosphorylated guanosine acids (ppGpp), γ-butyrolactones, small proteins such as σ-factors, as well as regulatory proteins encoded by regulatory genes biosynthetic gene clusters.”

Comment 2: Line 64: Provide basic information for PAS-LuxR family, including what PAS and LuxR stand for and how this type of regulator controls the transcription of the target genes (molecular mechanism of regulation). The authors provide related information in lines 121-127, but a clear explanation should be added in the Introduction. It would be useful if the authors introduced the knowledge from detailed studies on the role of this type of transcriptional regulator. What is the difference of PAS-LuxR from other types of regulators? This is an important point for the authors to shed light on this type of regulator.

Response 2: The basic information for PAS-LuxR family had been provided and this part had been revised to be “PAS domain, a signal module that monitors changes in light, redox potential, oxygen energy level of a cell, and small ligands, was first found in eukaryotes, and were named after homology to the Drosophila period protein (Per), the aryl hydrocarbon receptor nuclear translocator protein (ARNT) and the Drosophila single-minded protein (Sim). The PAS-LuxR family regulators are similar to other family in regulating secondary metabolic processes by influencing the transcription of functional genes. However, due to the presence of PAS, this regulator is more likely to play a regulatory role by sensing extracellular environmental changes. In addition, in terms of regulatory mode, the binding site of PAS-LuxR regulators is a 16 bp palindrom-like region, adjusts to the consensus CTVGGGAWWTCCCBAG (V represents A, C or G; W stands for A or T; B stands for C, G or T), located in -35 region in promoters of the regulated genes. ”

Comment 3: Figure 2. It would be useful if the localization of the promoter(s) controlled by the PAS-LuxR-type regulator is indicated in this figure. The meaning of the three different scale bars (upper, right) was not clear to this reviewer.

Response 3: According your suggestion, the localization of the promoter(s) controlled by the PAS-LuxR-type regulator is indicated in this figure. As the size of PKS genes were in the range of 5-30kb, the other genes were in the range of 0.2-2kb, the same scale bars would led the PKS genes too long or other genes too short. We use the different scale bars to make sure the gene clusters are clear.

Comment 4: Line 143: There is no indication of corresponding panel designations (a and b) in Fig. 3.

Response 4: The indication of corresponding panel designations (a and b) in Fig. 3 had been added.

Reviewer 2 Report

The authors provided a relatively thorough review of the PAS-LuxR family regulators in natural product biosynthesis. Please check the comments below:

  1. The title needs to be corrected, not ‘mentalism’.

  2. For figure 1, please draw all structures in publishable format. For example, the bond lengths look different, many bonds are not connected to the heavy atoms (such as compound FP), the angles are not all correct.

  3. For figure 2, please group each cluster based on their product type and similarity. Please fix the gene length bar, it is not clear what the author means by listing 3 bars. 

  4. For figure 3, please remove the annotation for natural amino acids since it is not necessary.

  5. For figure 4, please color the beta-scaffold, PAS core, Helical connectors into different colors since it is pretty confusing at this point.

  6. For Figure 6, please color the branches based on compound types. Please also include which compound it produces.  

  7. For figure 8, please color the PAS motif clearly.

  8. For figure 9, I think the authors mean ‘conserved sequence’ not ‘consistent sequence’

  9. Please add references to all the locations you failed to add, such as page 10, 2 graphs before section 5 (conclusion).

  10. Please add comments on the reasoning on why PAS-LuxR family regulators are needed in the gene cluster and their effects to the host.

  11. Please add comments on if the PAS-LuxR family regulators have been tested for the biosynthesis gene cluster other than their native gene cluster, aka inter-gene cluster-usage. This would be an application for heterologous expression system construction.

Author Response

Response to Reviewer 2 Comments

Comment 1: The title needs to be corrected, not ‘mentalism’.

Response 1: The misspellings of “mentalism” in title had been corrected with “metabolism”.

Comment 2: For figure 1, please draw all structures in publishable format. For example, the bond lengths look different, many bonds are not connected to the heavy atoms (such as compound FP), the angles are not all correct.

 Response 2: The structures had been revised in published format in Figure 1.

Comment 3: For figure 2, please group each cluster based on their product type and similarity. Please fix the gene length bar, it is not clear what the author means by listing 3 bars. 

Response 3: The gene clusters had been grouped based on their product type. As the size of PKS genes were in the range of 5-30kb, the other genes were in the range of 0.2-2kb, the same scale bars would led the PKS genes too long or other genes too short. We use two different scale bars to make sure the gene clusters are clear.

Comment 4: For figure 3, please remove the annotation for natural amino acids since it is not necessary.

Response 4: Based on your comment, the annotation for natural amino acids had been removed.

Comment 5: For figure 4, please color the beta-scaffold, PAS core, Helical connectors into different colors since it is pretty confusing at this point.

Response 5: Because secondary structures of N-terminal PAS structure are simulated by SWISS-MODEL, the secondary structures cannot be colored during generation on https://swissmodel.expasy.org/. We have arrows pointing out the corresponding positions.

Comment 6: For Figure 6, please color the branches based on compound types. Please also include which compound it produces.  

Response 6: The branches had been colored based on compound types, and the compounds were added in Figure 6. Please also include which it produces.

Comment 7: For figure 8, please color the PAS motif clearly.

Response 7: Because secondary structures of N-terminal PAS structure are simulated by SWISS-MODEL, the secondary structures cannot be colored during generation on https://swissmodel.expasy.org/. We marked the PAS motif with boxes.

Comment 8: For figure 9, I think the authors mean ‘conserved sequence’ not ‘consistent sequence’

Response 8: As the statement of reference, it has been revised as “Figure 9. Sequence logo comparison of TtmRIV and PAS-LuxR family regulon binding sites. a. Sequence logo of TtmRIV binding sites; b. Sequence logo of PAS-LuxR family regulator binding sites reported in the literature.”

Santos-Aberturas, J.; Vicente, C.M.; Guerra, S.M.; Payero, T.D.; Martin, J.F.; Aparicio, J.F. Molecular control of polyene macrolide biosynthesis: direct binding of the regulator PimM to eight promoters of pimaricin genes and identification of binding boxes. J Biol Chem 2011, 286, 9150-9161, doi:10.1074/jbc.M110.182428.

Cheng, Z.; Bown, L.; Tahlan, K.; Bignell, D.R. Regulation of coronafacoyl phytotoxin production by the PAS-LuxR family regulator CfaR in the common scab pathogen Streptomyces scabies. PLoS One 2015, 10, e0122450, doi:10.1371/journal.pone.0122450.

Comment 9: Please add references to all the locations you failed to add, such as page 10, 2 graphs before section 5 (conclusion).

Response 9: The references had been cited in their positions.

Comment 10: Please add comments on the reasoning on why PAS-LuxR family regulators are needed in the gene cluster and their effects to the host.

Response 10: The comments on the reasoning on why PAS-LuxR family regulators are needed in the gene cluster and their effects to the host had been added in the part of 5. Conclusions as “In the regulatory system, the PAS-LuxR regulators locate in the core position and have the potential role of corresponding extracellular environment changes, which increase the complexity and accuracy of the regulatory system to ensure the normal metabolism of Streptomyces.”

Comment 11: Please add comments on if the PAS-LuxR family regulators have been tested for the biosynthesis gene cluster other than their native gene cluster, aka inter-gene cluster-usage. This would be an application for heterologous expression system construction.

Response 11: The inter-gene-cluster-usage of PAS-LuxR family regulators was added in the PART of “4. Application of PAS-LuxR family regulators” as “FscRI is a necessary element for the heterologous expression of antimycin biosynthetic gene clusters in S. coelicolor. The heterologous expression of AmphRIV, NysRIV, or PteF can restore the production of pimaricin in pimM deletion strains. The expression of PimM can increase the yield of amphotericin and filipin in S. nodosus and S. avermitilis, respectively. The expression of pimM in pteF deletion strain also restored the production of filipin [30].”

Reviewer 3 Report

The review article is good and comprehensively describes the PAS-LuxR family regulator. However, some information should include improving this article:

1.      Is it the correct title? The highlighted phrase. Please refer back to the article.

"Effect of PAS-LuxR family regulators on the secondary mental- 2 ism of Streptomyces"

2.      There needs to be a method section for describing the gathering of information. Which platforms were used to obtain the reference articles? What are the keywords used for searching the article? What is the duration of gathering information?

3.      The word "actinobacteria" appeared in the sentence (44) "Approximately 59% of the regulators in the Actinobacteria contain an additional region, and most of the regulators in Streptomyces belonged to this type."

However, there needs to be more information about these previously. The author should add information about Actinobacteria before this statement.

   4. It is suggested to add current references (2021 and 2022) if any.

Author Response

Response to Reviewer 3 Comments

Comment 1: Is it the correct title? The highlighted phrase. Please refer back to the article. "Effect of PAS-LuxR family regulators on the secondary mentalism of Streptomyces"

Response 1: The misspellings of “mentalism” in title had been corrected with “metabolism”.

Comment 2: There needs to be a method section for describing the gathering of information. Which platforms were used to obtain the reference articles? What are the keywords used for searching the article? What is the duration of gathering information?

 Response 2: PAS-LuxR family, regulator, and secondary metabolite are selected as the keywords to search information on the Google Scholar from 2007 to 2022.

Comment 3: The word "actinobacteria" appeared in the sentence (44) "Approximately 59% of the regulators in the Actinobacteria contain an additional region, and most of the regulators in Streptomyces belonged to this type." However, there needs to be more information about these previously. The author should add information about Actinobacteria before this statement.

Response 3: In order to be clear and concise, we have revised the sentence to be “Most of the regulators in Streptomyces contain an additional region”

Comment 4: It is suggested to add current references (2021 and 2022) if any.

Response 4: We re-searched the literature and found that there was a new article in 2022, which we added it to the part of “3.3. Regulatory mechanisms of PAS-LuxR family regulators” as “In addition, PteF affected the transcriptions of genes related to various metabolic processes, including genetic information processing; DNA, energy, carbohydrate, and lipid metabolism; morphological differentiation; and transcriptional regulation, among others, but were particularly related to 10 potential secondary metabolites.”

Payero, T.D.; Rodríguez-García, A.; Barreales, E.G.; Pedro, A.d.; Santos-Beneit, F.; Aparicio, J.F. Modulation of Multiple Gene Clusters’ Expression by the PAS-LuxR Transcriptional Regulator PteF. Antibiotics 2022, 11, 994. https://doi.org/10.3390/antibiotics11080994